# Exogenous hCG Reduces Fetal Losses and Increases Litter Weight in Rangeland Goats During FTAI Protocol

**DOI:** 10.3390/ani15182704

**Published:** 2025-09-15

**Authors:** Jorge A. Bustamante-Andrade, Cesar A. Meza-Herrera, Oscar Angel-García, Ma Silvia Castillo-Zuñiga, Amaury Esquivel-Romo, Angeles De Santiago-Miramontes, Silvestre Moreno-Avalos, Martín Alfredo Legarreta-González, Viridiana Contreras-Villarreal, Francisco G. Véliz-Deras

**Affiliations:** 1Unidad Laguna, Universidad Autónoma Agraria Antonio Narro, Torreón 27054, Coahuila, Mexico; abaj_86@hotmail.com (J.A.B.-A.); angelgarciao@hotmail.com (O.A.-G.); mscz1970@hotmail.com (M.S.C.-Z.); angelesdesantiago867@gmail.com (A.D.S.-M.); moravasil70@gmail.com (S.M.-A.); 2Facultad de Agricultura y Zootecnia, Universidad Juárez del Estado de Durango, Venecia 35111, Durango, Mexico; amauryer@hotmail.com; 3Unidad Regional Universitaria de Zonas Áridas, Universidad Autónoma Chapingo, Bermejillo 35230, Durango, Mexico; cmeza2020@hotmail.com; 4Centro de Bachillerato Tecnológico Agropecuario No. 1, Torreón 27410, Coahuila, Mexico; 5Universidad Tecnologica de la Tarahumara, Guachochi 33180, Chihuahua, Mexico; mlegarreta@uttarahumara.edu.mx; 6Posgraduate Department, Fatima Campus, University of Makeni (UniMak), Makeni City 00232, Sierra Leone

**Keywords:** goats, reproduction, fetal losses, embryonic efficiency

## Abstract

The study aimed to evaluate the effects of administering two different doses of human chorionic gonadotropin (hCG) at two different times after fixed-time artificial insemination (FTAI) on early fetal loss and total litter weight in goats during the reproductive transition period. The research involved 40 crossbred multiparous goats, which were divided into five experimental groups. The results showed that administering 300 IU of hCG 14 days post FTAI significantly reduced early fetal losses, improved embryonic efficiency, and increased total litter weight. This suggests that using hCG in this manner can be an effective reproductive strategy for improving reproductive success in goats managed under extensive rangeland conditions.

## 1. Introduction

Extensive goat production systems have been established worldwide in arid and semi-arid zones with low vegetative capacity, where goats are generally managed under grazing conditions [1]. Reproductive performance in these production systems is a fundamental component in milk and kid production [2,3]. In Mexico, during the last decade, approximately 5 million goats have been found in extensive systems in arid and semi-arid zones [1,4]. In the Comarca Lagunera, located north of Mexico, the goat inventory exceeds 390,000 [5]. Goats from this region are classified as seasonal polyestrous, and, because of this, there is also seasonality in the production of milk, kids, and derivatives [6,7]: there is a period of deep anestrus from March to May and a breeding season that begins in August and ends in February, with a reproductive transition stage from June to July [8,9]. Hormonal strategies, such as the use of hCG, have been employed to counteract the effects of deep seasonal anestrus and the reproductive transition in the region [10,11]. The administration of hCG in different doses (example: 600, 500, 300, 100, and 50 IU) directly stimulates the dominant ovarian follicles in advanced stages of development, triggering ovulation, formation of the *corpus luteum*, an increase in the concentration of progesterone, and embryonic efficiency in goats in anestrus and reproductive transition, improving embryonic implantation [12,13,14]. Most pregnancy losses in goats occur during the preimplantation stage, making it a critical period that determines reproductive success [15]. Thus, failure in early pregnancies in this species is primarily caused by a deficiency in luteal function, which consequently affects early embryonic development [16,17,18]. In addition, asynchrony between the embryo and the uterus causes insufficient development, which is manifested by a reduction in the signaling of the embryo for the establishment of gestation and a decrease in the luteotrophic effect [19,20], concluding with the physiological process of embryo implantation [21,22]. In a recent study in the Comarca Lagunera, a high dose of hCG (300 IU) administered 14 days after Fixed-Time Artificial Insemination (FTAI) was effective in promoting luteogenesis and embryonic efficiency in goats during seasonal anestrus managed under an extensive production system [11]. Therefore, in this study, we hypothesize that administering 300 IU of hCG after FTAI will decrease early fetal loss and increase the embryo implantation rate and total litter weight in goats managed under extensive rangeland conditions during the reproductive transition period.

## 2. Materials and Methods

### 2.1. Ethics and Animal Welfare

All experimental procedures, methods, and handling of the experimental units used in this study complied with international [23] and national [24] guidelines for the ethical use, care, and welfare of research animals. In addition, institutional approval was obtained for its realization (UAAAN-UL-059/22).

### 2.2. Climatic Conditions of the Experimental Area

The study was carried out in the arid North of Mexico, in the Comarca Lagunera region (25° N, 103° W, altitude = 1111 m). This region has a dry climate, with an average annual temperature of 21 °C (37 °C from May to August, during summer, and 0 °C from November to February, during winter). Rainfall occurs from June to September with an annual average of 266 mm. The average relative humidity is 24% during the dry months and can rise to 78% during the rainy season [25]. In the Comarca Lagunera, goat production predominates under the sedentary extensive rangeland system; the goats feed on the native vegetation and agricultural waste [26].

### 2.3. Goats Handling Conditions

The goat management corral is in the goat farmer’s house. The goats were milked manually once a day (7:00 a.m.) and then released to their grazing routes (10:00 a.m. to 6:00 p.m.). Forty-five days before starting the experimental phase of this study, the herd was subcutaneously dewormed (Ivermectin 1%, Baymec, Bayer, Mexico City, Mexico) and fortified with vitamins A (500,000 IU), D3 (75,000), and E (50 mg) and Vigantol (ADE + Selenium 250 mL, Zapopan, Jalisco, Mexico). One month before the start of the study, water, shade, and mineral salts (17% P, 3% Mg, 5% Ca and 75% NaCl) were made available, with the goats given free access.

#### 2.3.1. Goat Females

Selection of goats. From a commercial herd of multiracial-crossbred goats (n = 155), 48 multiparous goats, with 45.3 ± 1.42 kg body weight, a body condition score of 1.96 ± 0.10, and 2–4 lactations, were selected and identified with earrings for better management during the experimental period.

Estrus induction protocol. Ovarian activity was determined in June by transrectal ultrasonography (5.3–10 MHz color Doppler equipment, Chison ECO-5, with a 12-inch probe). Each goat underwent ultrasound scanning on days 14 and 7 before the application of hCG to confirm the presence of corpora lutea in both ovaries. Goats (13/48 = 27.08%) that had functional corpora lutea were given 1 mL of an analog of prostaglandin F_2_ alpha (D-Cloprostenol^®^ Sanfer, i.m., Ciudad de México, Mexico) to induce lysis of the corpus luteum. Once the absence of corpora lutea was confirmed, the goats were subjected to an estrus induction protocol. On day −1, all females received 20 mg of P4 (Progesvit^®^, Brovel, i.m., Irapuato, Guanajuato, Mexico) to prevent short cycles. On day zero (0 d), the goats received 200 IU of i.m. hCG (Chorulon^®^, Intervet, Mexico City, Mexico) to stimulate the formation of the antrum in the ovarian follicles in advanced stages of development and, in turn, promote ovulation. The morning of the next day, for the detection of estrus, three sexually active adult male goats, provided with aprons to avoid copulation with the females, were used. Of the total number of goats (n = 48) exposed to the induction protocol, 40 responded, exhibiting estrus.

#### 2.3.2. Male Goats, Extraction Process, Macro and Micrometric Semen Evaluation

The male goats (n = 3; Grenadine breed; 2.5 years old), used for semen collection, were subjected to treatment with an application of 50 mg of testosterone (Testosterone-50, androgenic steroid, i.m., Lab. Brovel, Mexico City, Mexico) every third day for three weeks before the start of the study period to improve spermatogenesis [27]. At the end of June, semen was extracted using an artificial vagina (Walmur-Veterinaria, Montevideo, Uruguay); the ejaculate was evaluated, and only ejaculates with volume ≥ 0.5 mL, sperm concentration ≥ 2500 × 10^6^ cells/mL, and sperm mass motility ≥ 3 (scale 0–5) and progressive motility ≥ 70% were used. To avoid any effect on the embryo implantation rate, the collected semen samples were mixed and subsequently diluted using a commercial diluent, following the instructions provided by the laboratory (OptidylTM, Cryo-Vet, León, Guanajuato, Mexico).

### 2.4. Fixed-Time Artificial Insemination

All goats (n = 40) were exposed to the FTAI protocol 48 h after the application of hCG. The insemination procedure was carried out using a vaginoscope (Walmur-Veterinary, Montevideo, Uruguay) equipped with a light source. The semen was deposited in the pericervical area, and two services were provided in the morning (9:00 a.m.) and afternoon (7:00 p.m.). All females were inseminated twice, 40 and 48 h after administering human chorionic gonadotropin.

### 2.5. Conformation of the Treatments

After inseminating the study goats (n = 40), they were distributed into five experimental treatments, taking into account two doses of hCG (100 and 300 IU, Chorulon^®^, Intervet, Mexico City, Mexico) and two application times (7 and 14 d post-AITF), plus a control group. The treatments were as follows: (1). G100-7 (n = 8), 100 IU of hCG, 7 days after AITF; (2). G100-14 (n = 8), 100 IU of hCG, 14 days after AITF; (3). G300-7 (n = 8), 300 IU of hCG, 7 days after AITF; (4). G300-14 (n = 8), 300 IU of hCG, 14 days after AITF, and (5). CONT (n = 8), 0.5 mL of saline solution, 7 and 14 days after IATF.

### 2.6. Response Variables

#### 2.6.1. Body Weight, Body Condition, and Estrus Induction Protocol

Body weight and body condition were measured at the start of fieldwork. Body weight (BW) was determined using a digital scale with a capacity of 250 kg and a precision of 50 g (Torrey 110v/220v, Jalisco, Mexico). BCS was also determined by an experienced technician, as described by Walkden-Brown et al. [28], using a scale of 1 (very thin) to 4 (very fat). Once anovulation was determined, the goats were subjected to the estrus induction protocol mentioned above.

#### 2.6.2. Ovulation Rate, Corpus Luteum Diameter, Luteal Area, Embryo Implantation Rate, and Embryonic Efficiency Indices

The percentage of females that ovulated was determined on days 0 and 10 after the estrus induction protocol was initiated. It was calculated based on the observation of corpora lutea (CL) using transrectal ultrasound evaluation (5.3–10 MHz, Chison ECO-5 color Doppler equipment, with a 12-inch probe). The ovulation rate (OVR) was calculated by observing CL through ultrasound on day 10 after FTAI. In addition, the luteal area (CLA) was determined by measuring the diameter of the corpus luteum (CLD). On day 30 post FTAI, the embryo implantation rate (EIR) was determined by the same route. In addition, two indices were developed to weigh the success of the embryo implantation rate, concerning the conception rate and the fecundity rate: embryonic efficiency index 1 [IEE1 = (embryo implantation rate) (conception rate/100)] and embryonic efficiency index 2 [IEE2 = (embryonic implantation rate) (fecundity rate/100)].

#### 2.6.3. Rates of Conception, Fertility, Fecundity, and Prolificacy

These response variables were determined as follows: the conception rate (CR) was determined on day 45 post insemination, considering the number of pregnant goats/number of inseminated goats; for the fertility rate (TFR), the number of pregnant females that gave birth was considered; for the fecundity rate (FCR), the number of fetuses per inseminated goat was considered; and the prolificacy rate (PR) was determined at parturition, considering the number of offspring born per pregnant goat.

#### 2.6.4. Early Fetal Losses, Birth Weight, and Total Litter Weight

Fetal loss percentages at day 30, day 45 post FTAI, and until delivery were determined from transrectal ultrasonography. The birth weight of the kids was recorded immediately after the mother finished cleaning them to avoid disturbing the mother–kid bond. For birth weight measurement, a scale with a capacity of 40 kg and a precision of 5 g (Torrey LPCR 40 USB, 40 kg/5 g, port USB A/B, Jalisco, Mexico) was used. The total weight of the litter was determined by multiplying the total number of kids by their weight.

Therefore, the potential effects of these five experimental treatments on early fetal losses, embryonic efficiency, litter size, and other reproductive variables are illustrated in Figure 1.

### 2.7. Statistical Analysis

A first linear model was developed to assess the potential relationship between the hCG dose (i.e., 100 vs. 300 IU) and day of administration (i.e., 7 vs. 14 days after FTAI) concerning body weight (BW, kg), total litter weight (kg), the diameter of the corpus luteum (CLD, mm), and the area of the corpus luteum (CLA, mm^2^). Regarding percentage and count variables, the body condition score (BCS, units), estrus induction (EI, %), conception rate (CR, %), fertility rate (FR, %) prolificacy rate (PR, %), fecundity rate (FCR, %), ovulation rate (OVR, units), number of embryos (EN, units), embryo implantation rate (EIR, %), embryonic efficiency index 1 (IEE-1, %), embryonic efficiency index 2 (IEE-2, %), and fetal losses at d 30, d 45, d 30–45, and d 45, at delivery, and in total (%) were log10-transformed before performing ANOVA to overcome skewness, as the data did not fit the normal distribution. Least squares means and standard errors were computed for each experimental treatment, and multiple comparisons of means were performed using Fisher’s LSD–LSMEANS option of the SAS PROC GLM. Since all experimental treatments were evaluated individually, each goat within the experimental group was defined as an experimental unit. Differences in treatments were considered statistically significant if *p* < 0.05. All analyses were calculated using SAS procedures (SAS Inst. Inc., Version 9.4, Cary, NC, USA).

## 3. Results

### 3.1. Body Weight, Body Condition Score, and Estrus Induction

These variables did not show statistically significant differences (*p* > 0.05) between the experimental groups. In general, the average body weight was 45.34 ± 1.42 kg, the body condition score was 1.96 ± 0.10 units, and estrus induction was achieved in 83.5% of the animals (Table 1).

### 3.2. Ovulation Rate, Corpus Luteum Diameter, Luteal Area, Embryo Implantation Rate, and Embryonic Efficiency Indices

The higher embryo implantation rate and the largest corpus luteum area (*p* < 0.05) favored the experimental group G300-14 (Table 2), which was the highest among all groups across all variables. However, although there were no statistical differences (*p* > 0.05) between groups G100-7, G100-14, G300-7, and CONT for both efficiency indices (EEI1 and EEI2), the higher values for the index favored (*p* < 0.05) G300-14 compared to the other treatments.

### 3.3. Conception, Fertility, Prolificacy, and Fecundity Rates

There was no effect of the two doses of hCG (100 vs. 300 IU) at different times (7 vs. 14 d) after FTAI concerning CONT (Table 3). However, the variables of conception rate, fertility rate at delivery, and fecundity rate showed significant statistical differences (*p* = 0.04, *p* = 0.05, *p* = 0.03) favoring the experimental group G300-14.

### 3.4. Early Fetal Losses, Birth Weight, and Total Litter Weight

There was no statistical difference (*p* > 0.05) between the experimental groups and the CONT group regardless of the type of delivery (Table 4); however, G300-14 (nine kids) was the one that presented the most significant number of offspring, including single births, twins, and the only triple birth, while the rest of the groups had fewer kids: G100-7 (four kids), G100-14 (three kids), G300-7 (four kids), and CONT (two kids). Finally, it is worth noting that the variable of total weight of the litter showed a significant difference in favor of the group G300-14 (*p* < 0.05), with a difference of approximately 20 kg between the best group and the control.

## 4. Discussion

The results obtained in this study support our hypotheses, which establish that a high dose of hCG (300 IU) is effective in reducing early fetal losses, as well as enhancing embryonic efficiency and litter size. In this sense, the highest values of the response variables, conception rate (CR), fertility rate (FR), fecundity rate (FCR), *corpus luteum* area (CLA), embryo implantation rate (EIR), and embryonic efficiency indices 1 and 2 (IEE1 and EEI2), were observed in G300-14 with the administration of a high dose of hCG (300 IU) two weeks (14 d) after the application of the protocol. The findings of this study coincide with what was previously reported by [11], who evaluated the effect of hCG doses post FTAI in goats in an extensive sedentary system during deep seasonal anestrus (April, 25° LN) in the Comarca Lagunera and found better results with the high dose of hCG on luteogenesis and embryonic efficiency. This is surely due to the dual effect of this exogenous hormone, which coincides with the function of gonadotropins LH and FSH and consequently promotes the better development of ovarian structures [29,30], promoting the process of embryonic establishment in a tripartite synchrony between the embryo, the maternal recognition of gestation, and the receptivity of the uterus, thus reducing fetal losses [31], as observed in this study, where we found statistically lower values of fetal loss at d 30, at d 45, and overall, favoring G-300-14 compared to the rest of the groups. Additionally, the highest total weight of the litter was observed in this experimental group. In effect, this research helps mitigate the adverse effects of early fetal loss, thereby increasing total litter weight by utilizing hormonal protocols in female goats during the reproductive transition stage, which promotes better physiological and endocrine conditions during the critical period of gestation.

At the beginning of the experimental period, 83.33% of the goats (40/48) responded favorably to the estrus induction protocol. In various studies in the Comarca Lagunera, similar induction protocols have been used, with 20 mg of P4 plus hCG in different doses, for example, as reported by Alvarado-Espino et al. [9,10], who used different doses of hCG (0, 50, 100 and 300 IU) in goats during the early reproductive transition period (June) and reported an effect close to 90% for females in estrus. Regarding the variables of body weight (BW) and body condition score (BCS), they were not different between the experimental groups and the CONT group because the goats were selected based on the homogeneity of these variables to ensure group homogeneity.

Regarding the response variables of ovulation rate (OVR), *corpus luteum* diameter (CLD), and number of embryos (EN), they were not different between treatments, most likely due to the effect of the reproductive transition period; that is, the goats required less energy for these physiological processes. In this sense, these results are consistent with those found in local goats of the Comarca Lagunera in the period of early reproductive transition [9,13] for the same variables; however, the embryo implantation rate was different between the treatments, favoring the group that received the high dose of hCG, G300-14, promoting the highest embryonic efficiency against the rest of the treatments. These results coincide with what was reported in different studies in small ruminants, where higher embryo implantation rates were found [32,33,34]. A possible explanation for such a lack of effect in the 100 IU groups could be related to the extremely high sensitivity of the hypothalamus–pituitary axis to negative gonadal E2 feedback experienced by goats facing a reproductive transition period; i.e., coming from a seasonal anestrus, most hormonal actions depend on certain thresholds to elicit a definite response. It is interesting to mention that in our study, the largest luteal area (CLA) favored the G300-14 group, coinciding with an investigation in which they evaluated 300 IU of hCG 7d post estrus in Toggenburg goats [35] under an intensive system (21° South), observing a positive relationship between the luteal area and the high dose of hCG; likewise, 11, reported similar findings in local goats from the Comarca Lagunera in seasonal anestrus, treated with different doses of hCG in a marginal production system. This information aligns with that reported by Rodrigues et al. [36], who evaluated the effect of intravaginal hCG on ovarian function in dairy goats and found a larger luteal area. To assess the success of embryo implantation rates in relation to conception and fertility rates, our study established two embryonic efficiency indices: IEE1 and IEE2. The values generated by both indices favored the G300-14 group, increasing the process of maternal recognition of pregnancy, for which we established that a high dose of hCG at this physiological stage is necessary for greater synthesis and function of IFN [18,37,38], for example, counteracting the antagonistic effect of prostaglandin F2a to prevent regression of the *corpus luteum* [15].

Regarding the response variable prolificacy rate (PR), it did not differ between the experimental groups of this study; this information coincides with that reported by Rodríguez-Martínez et al. [39], who found a prolificacy rate of 1.55 ± 0.02 in a study where hCG was used in goats from the Comarca Lagunera. The above differs from that reported by Martins et al. [40], who evaluated the luteal characteristics and serum progesterone concentrations in dairy goats subjected to intramuscular and intrauterine hCG and found a positive effect on the prolificacy rate; however, the highest values for the variables of conception rate (CR), fertility rate (FR), and fecundity rate (FCR) favored the G300-14 group, meaning that the proportion of the number of fetuses with respect to the number of inseminated females was influenced by the effect of the high dose of hCG at 14 d post IATF. These results agree with Fonseca et al. [41] and Fernandez et al. [20], who carried out a study where the conception rate was similar, using hCG in addition to other exogenous hormones at different doses in goats in Brazil. In another study conducted on Merino sheep, GnRH (4 µg i.m.) and hCG (300 IU i.m.) were administered on day 4 post FTAI in northern Patagonia, yielding the best results for fertility and fecundity rates. Success in the gestation of goats, as in other domestic ruminants, is conditioned by the adequate functionality of the *corpus luteum* for the release of progesterone [15,32,34]. This hormone is closely related to the formation and functioning of interferon-tau (INFt) for the process of early recognition of pregnancy, together with prostaglandin and oxytocin [15,18,19,38,42,43].

Regarding the variables related to early fetal loss, our research found that the best results were obtained by the G300-14 group, which did not experience early fetal loss during the first 30 and 45 days after FTAI and reported only one fetal loss in total. Unlike the rest of the treatments, these results are in line with what was reported by Fernandez et al. (2019) [20], who carried out a study on Merino sheep, administering GnRH (4 µg i.m.) and hCG (300 IU i.m.) at day 4 post FTAI in northern Patagonia, and found a statistical difference in the lower number of early fetal losses (d 33 post IATF) for the group of sheep in which hCG was used. Early fetal losses in small ruminants are estimated to range from 8% to 30% before day 30 of gestation [44,45]. These early losses coincide with the expansion of the concept and the process of placentation. In recent studies, it was shown that treatment with hCG, used in the early luteal phase of sheep, showed a positive effect in reducing early fetal losses, probably related to the formation of accessory *corpora lutea*; in this sense, Bartolome et al. [46], and Garcia-Pintos & Menchaca [47], established that the luteotrophic action of hCG produces an increase in the concentration of progesterone necessary for an adequate intrauterine environment to improve embryonic survival and therefore reduces fetal losses in ruminants.

Regarding the weight of the kids at birth, there was no statistical difference between the experimental groups and the CONT group, regardless of the type of delivery. However, G300-14 was the group that presented the largest number of kids, including single births, twins, and the only triple birth. Regarding the total weight of the litter, the G300-14 group exhibited the highest weight, indicating a positive effect of the high dose of hCG on this response variable. These results can be extrapolated to the number of offspring in the goat production system in the Comarca Lagunera, which represents a greater source of income for producers. In this sense, research conducted in goat production systems under marginal conditions has reported an incidence of up to 70% of fetal losses in mestizo goats in Mexico, affecting the global prolificacy of the herds [3]. In another study conducted by Fernandez et al. [20], a larger litter size was reported in Merino sheep under extensive grazing conditions when hCG was applied (n = 38 kids) compared to GnRH (n = 28 kids) during the breeding season in Argentina.

The proposal we put in place with the use of a high dose of hCG in goats during the reproductive transition season represents an alternative solution to the problem of early fetal loss, reduced embryonic efficiency, and reductions in total litter weight, which goat breeders face in extensive production systems, translating into substantial economic losses. Since goats require less metabolic demand at this time of year to restart their reproductive cycle, we prioritize the use of reproductive biotechnology, such as FTAI, and the administration of hCG in extensive goat production systems to increase the reproductive efficiency of herds through genetic improvement. However, it is a priority to continue scientific research on this species and under this production system, as they are a significant source of income for the global population, especially in marginal regions.

It is essential to highlight that, as observed in the literature reviewed, most studies conducted both in this region [9,10,12,17,39] and globally [11,13,20,29,30,40,41,46] have focused on reproductive and productive variables such as prolificacy rates. In contrast, the present study incorporates additional outcome measures, specifically the reduction in fetal losses during the embryonic implantation period and the total litter weight, thereby broadening the scope of applied and academic research impacts. The incorporation of these novel variables provides significant practical value. By employing this alternative reproductive control strategy, goat producers stand to benefit from increased incomes, particularly by ensuring milk and kid production during periods of limited supply, which in turn allows for improved market prices for both milk and kids. Moreover, when the findings of this study are considered alongside the total number of offspring born per season in the region, the results point to substantial opportunities for further inquiry—such as exploring the efficacy of lower hCG doses—and the development of targeted scientific questions.

## 5. Conclusions

Our results confirm that a high dose of hCG (300 IU) two weeks (14 d) after FTAI helps reduce early fetal losses in goats during the reproductive transition period and, by inherent consequence, elicits a greater embryonic efficiency, coupled with an increase in the total weight of the litter, considering that the best results in fecundity rate, conception rate, *corpus luteum* area, and embryonic efficiency indexes 1 and 2, in addition to the lowest fetal losses, favored the experimental group G300-14. With the completion of this study, one more alternative is ratified for the increase in productivity in goat production systems using the hormone hCG as a reproductive strategy to reduce embryonic mortality during the critical period of gestation, in the reproductive transition season, due to the lower energy demand to restart the reproductive cycles in this species.

## Figures and Tables

**Figure 1 animals-15-02704-f001:**
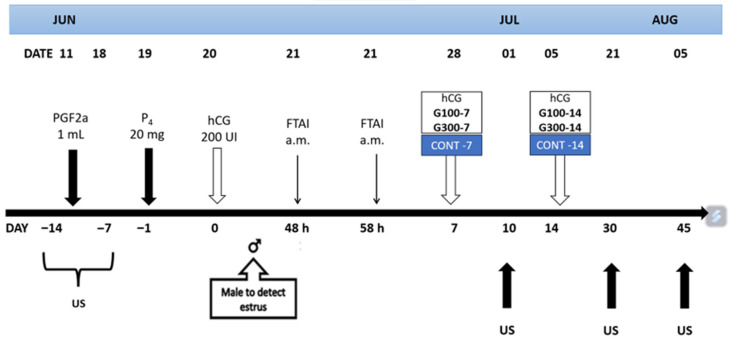
Schematic representation of the experimental protocol. The application of hCG (100 or 300 IU) was on day 7 or 14 post FTAI. Transrectal ultrasonography was performed to determine the ovulatory status (US1 and US2), ovulatory rate (US3), embryo implantation rate (US4), and conception rate (US5) in goats kept under an extensive system and in reproductive transition (June, 25° N).

**Table 1 animals-15-02704-t001:** Least squares means ± standard error of body weight (BW), body condition score (BCS), and estrus induction (EI) of multi-breed and multiparous goats (n = 40, Alpine, Saanen, Nubian × Criollo) managed under extensive conditions in northern Mexico, treated with either 100 or 300 IU of hCG or saline solution at 7 or 14 days post Fixed-Time Artificial Insemination, during the reproductive transition season (June, 25° N).

Variables	G100-7 (n = 8)	G100-14 (n = 8)	G300-7 (n = 8)	G300-14 (n = 8)	CONT (n = 8)	*p* Value
BW (kg)	45.6 ± 1.84	44.8 ± 1.14	45.2 ± 1.06	45.7 ± 1.60	45.4 ± 1.49	0.90
BCS (units)	2.0 ± 0.09	1.9 ± 0.07	2.0 ± 0.13	2.0 ± 0.08	1.9 ± 0.13	0.56
EI (n, %)	8/10 (80.0)	8/9 (88.8)	8/10 (80.0)	8/10 (80.0)	8/9 (88.8)	0.90

No differences (*p* > 0.05) for any variable occurred among experimental groups.

**Table 2 animals-15-02704-t002:** Least squares mean ± standard error of ovulation rate (OVR), *corpus luteum* diameter (CLD), *corpus luteum* area (CLA), number of embryos (EN), embryo implantation rate (EIR), embryonic efficiency index 1 (IEE-1), and embryonic efficiency index 2 (IEE-2) of multiracial and multiparous goats (n = 40, Alpine, Saanen, Nubian × Criollo) managed under extensive conditions in northern Mexico and treated with 100 or 300 IU of hCG or saline solution 7 or 14 days after Fixed-Time Artificial Insemination, during the reproductive transition season (June, 25° N).

Variables	G100-7 (n = 8)	G100-14 (n = 8)	G300-7 (n = 8)	G300-14 (n = 8)	CONT (n = 8)	*p* Value
OVR (n)	0.87 ± 0.33	1.1 ± 0.48	1.4 ± 0.17	1.5 ± 0.33	0.68 ± 0.17	0.46
CLD (mm)	10.6 ± 0.42	9.9 ± 0.45	10.9 ± 0.96	12.9 ± 0.72	8.68 ± 0.98	0.23
CLA (mm) ^2^	87.35 ± 0.18 ^b^	97.58 ± 0.32 ^b^	100.11 ± 0.45 ^b^	135.66 ± 0.25 ^a^	95.02 ± 0.36 ^b^	0.04
EN (n)	1.60 ± 0.18	1.80 ± 0.42	1.80 ± 0.20	2.0 ± 0.26	1.40 ± 0.20	0.59
EIR (n, %)	5/8 (62.5) ^b^	5/8 (62.5) ^b^	6/8 (75.0) ^b^	8/8 (100.0) ^a^	3/8 (37.5) ^b^	0.03
EEI ^1^ (%)	23.43 ± 0.25 ^b^	31.25 ± 0.21 ^b^	46.87 ± 0.21 ^b^	87.50 ± 0.20 ^a^	14.06 ± 0.24 ^b^	0.02
EEI ^2^ (%)	31.25 ± 0.33 ^b^	23.43 ± 0.21 ^b^	37.5 ± 0.29 ^b^	87.50 ± 0.33 ^a^	9.37 ± 0.29 ^b^	0.03

Different letters between columns show differences (*p* > 0.05). Data are presented as means and standard errors of the mean. Embryonic efficiency index ^1^ = [implantation rate] [conception rate/100]. Embryonic efficiency index ^2^ = [implantation rate] [fecundity rate/100].

**Table 3 animals-15-02704-t003:** Least squares means ± standard errors of conception rate (CR), fertility rate (FR), prolificacy rate (PR), and fecundity rate (FCR) of multibreed and multiparous goats (n = 40, Alpina, Saanen, and Nubian × Criollo), managed under extensive conditions in the north of Mexico and treated with 100 or 300 IU of hCG or saline solution at 7 or 14 days after Fixed-Time Artificial Insemination, during the reproductive transition season (June, 25° N).

Variables	G100-7 (n = 8)	G100-14 (n = 8)	G300-7 (n = 8)	G300-14 (n = 8)	CONT (n = 8)	*p* Value
CR (n, %)	3/8 (37.5) ^b^ ± 0.27	4/8 (50.0) ^b^ ± 0.14	5/8 (62.5) ^b^ ± 0.2	8/8 (100) ^a^ ±0.14	3/8 (37.5) ^b^ ± 0.1	0.04
FR (n, %)	3/8 (37.5) ^b^ ±0.09	3/8 (37.5) ^b^ ± 0.14	4/8 (50.0) ^b^ ± 0.1	7/8 (87.5) ^a^ ±0.25	2/8 (25.0) ^b^ ± 0.13	0.05
PR (n)	1.3 ± 0.23	1.5 ± 0.33	1.5 ± 0.33	2.0 ± 0.29	1.2 ± 0.29	0.25
FCR (n, %)	4/8 (50.0) ^b^ ± 0.2	3/8 (37.5) ^b^ ± 0.11	4/8 (50.0) ^b^ ± 0.1	9/8 (112.5) ^a^ ± 0.2	2/8 (25.0) ^b^ ± 0.06	0.03

^a,b^ Response variables with different superscripts within lines differ (*p* > 0.05).

**Table 4 animals-15-02704-t004:** Least squares means ± standard errors of fetal losses at d 30 and d 45, between d 30 and d 45, and from 45 to delivery and birth weight of kids from multiracial and multiparous goats (n = 40, Alpina, Saanen, and Nubia × Criollo) managed under extensive conditions in northern Mexico and treated with 100 or 300 IU of hCG or saline at 7 or 14 days post Fixed-Time Artificial Insemination, during the reproductive transition season (June, 25° N).

Variables	G100-7 (n = 8)	G100-14 (n = 8)	G300-7 (n = 8)	G300-14 (n = 8)	CONT (n = 8)	*p* Value
Fetal losses at d 30 post FTAI (%)	3/8 (37.5 ± 0.12) ^b^	3/8 (37.5 ± 0.25) ^b^	2/8 (25.0 ± 0.19) ^b^	0/8 (0.0) ^a^	5/8 (62.5 ± 0.13) ^b^	0.04
Fetal losses at d 45 post FTAI (%)	5/8 (62.5 ± 0.1) ^b^	4/8 (50 ± 0.13) ^b^	3/8 (37.5 ± 0.15) ^b^	0/8 (0.0) ^a^	5/8 (62.5 ± 0.18) ^b^	0.03
Fetal losses between days 30 and d 45 post FTAI (%)	2/8 (25 ± 0.12)	1/8 (12.5 ± 0.23)	1/8 (12.5 ±0.2)	0/8 (0.0)	0/8 (0.0)	0.53
Fetal losses between d 45 post FTAI and the birth (%)	0/8 (0.0)	1/8 (12.5 ± 0.25)	1/8 (12.5 ± 0.14)	1/8 (12.5 ± 0.1)	1/8 (12.5 ±0.25)	0.76
Total fetal loss (%)	5/8 (62.5 ±0.2) ^b^	5/8 (62.5 ±0.2) ^b^	4/8 (50 ± 0.10) ^b^	1/8 (12.5 ± 0.2) ^a^	6/8 (75 ± 0.1) ^b^	0.02
Birth weight of kids (kg)						
Single (n = 11)	3.8 ± 0.51 (2)	3.5 ± 0.32(1)	3.7 ± 0.42(2)	3.6 ± 0.5 (4)	3.5 ± 0.6 (2)	0.45
Twin (n = 8)	2.6 ± 0.15(2)	2.4 ± 0.23(2)	2.8 ± 0.56(2)	2.9 ± 0.63(2)	nd * (0)	0.65
Triple (n = 3)	nd * (0)	nd * (0)	nd * (0)	2.2 ± 0.36(3)	nd * (0)	-
Totals (n = 22)	(4)	(3)	(4)	(9)	(2)	0.10
Total litter weight (kg)	12.8 ± 0.31 ^b^	8.3 ± 0.25 ^b^	13 ± 0.6 ^b^	26.8 ± 0.23 ^a^	7 ± 0.10 ^b^	0.02

^a,b^: Different letters in the same row indicate *p* < 0.05 (statistical significance); * nd: no data.

## Data Availability

The original contributions presented in this study are included in the article. Further inquiries can be directed to the corresponding authors.

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
