# Peer review of "Exogenous hCG Reduces Fetal Losses and Increases Litter Weight in Rangeland Goats During FTAI Protocol"

_animals, 2025, doi:10.3390/ani15182704_

Round 1
Reviewer 1 Report
Comments and Suggestions for Authors
Notes and recommendations
- In the introduction, the authors should indicate why they chose this particular drug (hCG)? It is well known that the use of GnRH also reduces fetal losses.
- At the end of the introduction, the purpose of the present study should be clearly stated.
- Lines 124 bad 125 do not match Figure 1? Must be corrected.
- In material and methods: It is not stated whether the goats were tested for heat by teasers or estrus detector?
- It is unclear, how the authors calculated the embryo implantation rate - by calculating embryos or corpora lutea.
- Lines 246 and 247 are in Spanish. Must be corrected.
- Overall, the applied protocol for estrus synchronization is new (and in my opinion controversial) and the authors should describe it in more details in Discussion.
Author Response
Reviewer 1
Comment 1: In the introduction, the authors should indicate why they chose this particular drug (hCG)? It is well known that the use of GnRH also reduces fetal losses
Response to comment 1: We chose the hCG hormone because of its immediate effect in inducing estrus in the first instance, as well as because it gave us very good results in a previous study (2021). Furthermore, hCG has a similar structure to LH (70%) and has a longer half-life than LH (~39 h vs. 6 h). It also has 30% of the FSH hormone, which promotes ovarian stimulation at the level of theca cells and FSH receptors in granulosa cells. It is also important in reducing embryonic loss during the critical period of gestation.
Text in document for comment 1: Human chorionic gonadotropin (hCG) exhibits a structural similarity to luteinizing hormone (LH), with a 70% similarity in its molecular composition. The substance also contains 30% of the FSH hormone, which, by acting on theca cells and FSH receptors in granulosa cells, stimulates the ovaries. As Fernandez et al. (2019) demonstrate, this is also important in reducing embryonic loss during the critical period of pregnancy.
Comment 2: At the end of the introduction, the purpose of the present study should be clearly stated.
Response to comment 2: Added.
Text in document for comment 2: Therefore, the objective of this study was to evaluate the effect of administering 300 IU of hCG after AITF on early fetal loss and embryo implantation rate, as well as total litter weight in goats managed under extensive rangeland conditions during the reproductive transition period.
Comment 3: Lines 124 bad 125 do not match Figure 1? Must be corrected.
Response to comment 3: Corrected
Comment 4: In material and methods: It is not stated whether the goats were tested for heat by teasers or estrus detector?
Response to comment 4: The detection of estrus in goats following the induction protocol was carried out the morning after the administration of hCG, using three sexually active males fitted with protective devices to prevent penetration and ejaculation. Once the females in estrus were identified, they were separated for subsequent insemination, as outlined in the materials and methods section.
Text in document for comment 4: The following morning, three sexually active adult male goats were utilized for the detection of estrus. The goats were provided with aprons to avoid copulation with the female. Among the total number of goats exposed to the induction protocol (n = 48), 40 exhibited a response, manifesting estrus.
Comment 5: It is unclear how the authors calculated the embryo implantation rate - by calculating embryos or corpora lutea.
Response to comment 5: Corrected.
Comment 6: Lines 246 and 247 are in Spanish. Must be corrected.
Response to comment 6: Corrected.
Comment 7: Overall, the applied protocol for estrus synchronization is new (and in my opinion controversial) and the authors should describe it in more details in Discussion.
Response to comment 7: Added.
Text in document for comment 7: The estrus induction protocol utilized in this study has previously been evaluated in goats from diverse latitudes within the Comarca Lagunera region. A favorable response has been observed in the use of 20 mg of progesterone on day -1 to prevent short cycles and 200 IU of hCG on day 0 to promote ovulation, as previously described. This response can be attributed to the analogy of this hormone with LH, as previously discussed. The UAAAN UL team has conducted a series of studies that have yielded estrus induction rates exceeding 90%.
Reviewer 2 Report
Comments and Suggestions for Authors
Please check the following details that raised my uncertainity:
Line 65 and 70, it is not clear if the cited authors
Alvarado -Espino covers the article 2019a or b
Line 125: instead of On day 1 should be On day -1.
The title 2.6.1. doesnt.need Estrus induction part as it.is.already.explained.in details in chapter 2.3.1.
Line 247 and 248 are written in Spanish and should be translated in English.
Line 320 and 321 : citations should not bear parenthesis
Line 361: citation 38 is wrongly citated
Author Response
Reviewer 2
Please check the following details that raised my uncertainty:
Comment 1: Line 65 and 70, it is not clear if the cited authors Alvarado -Espino covers the article 2019a or b.
Response to comment 1: Reference corrected throughout the article.
Comment 2: Line 125: instead of “On day” 1 should be “On day -1”.
Response to comment 2: Corrected.
Comment 3: The title 2.6.1. doesn’t need Estrus induction part as it is already explained in detail in chapter 2.3.1.
Response to comment 3: As outlined in Section 2.3.1, the subject is mentioned exclusively within the context of response variables. A detailed description is not provided.
Comment 4: Line 247 and 248 are written in Spanish and should be translated in English.
Response to comment 4: Corrected.
Comment 5: Line 320 and 321: citations should not bear parenthesis
Response to comment 5: Corrected.
Comment 6: Line 361: citation 38 is wrongly citated
Response to comment 6: Corrected.
Reviewer 3 Report
Comments and Suggestions for Authors
This study investigates the effects of administering two different doses of human chorionic gonadotropin (hCG; 100 IU and 300 IU) at two time points (Day 7 and Day 14) following fixed-time artificial insemination (FTAI) on key reproductive outcomes in goats. Forty crossbred multiparous goats underwent estrus synchronization and FTAI and were randomly assigned to five experimental groups (n = 8 per group): G100-7 (100 IU hCG on Day 7), G100-14 (100 IU on Day 14), G300-7 (300 IU on Day 7), G300-14 (300 IU on Day 14), and a control group (0.5 mL saline on Days 7 and 14).
Evaluated parameters included corpus luteum (CL) area, embryo implantation rate, embryonic efficiency indices (EEI-1 and EEI-2), conception rate, fertility rate, fecundity rate, fetal loss at Days 30 and 45, and total litter weight. The G300-14 group exhibited superior reproductive performance, with reduced early fetal losses, enhanced embryonic efficiency, and greater total litter weight. The authors conclude that administering 300 IU of hCG during the transitional reproductive period is a beneficial strategy for improving reproductive efficiency, particularly in low-input goat production systems.
-
Lack of Nutritional Control Data
The manuscript does not provide any information regarding dietary composition or feed intake during the period from estrus synchronization to early pregnancy. Considering that nutritional status can significantly influence luteal function and embryonic survival, it is important to clarify whether all animals were managed under uniform feeding conditions. Inclusion of such data would strengthen the validity and reproducibility of the findings. -
Experimental Design Overlaps Significantly with Previous Literature
The experimental design is highly similar to that of a previously published study (Biology 2021, 10, 429), particularly with respect to hCG dosage, timing of administration, use of FTAI, and several key outcome measures such as CL area. Although this manuscript presents additional data—such as fetal loss rates and litter weight—the underlying mechanism investigated (i.e., enhancement of luteal function via hCG) remains essentially the same. As a result, the study lacks sufficient novelty and appears to be a confirmatory replication of existing work. The authors should clearly articulate the distinction between their findings and those of previous studies, and emphasize any original contributions. -
Outdated References
Several of the references cited are relatively dated. It is recommended that the authors update their literature review with more recent studies on the application of hCG in small ruminant reproductive management to ensure the manuscript reflects current knowledge and research trends.
Author Response
Reviewer 3
This study investigates the effects of administering two different doses of human chorionic gonadotropin (hCG; 100 IU and 300 IU) at two time points (Day 7 and Day 14) following fixed-time artificial insemination (FTAI) on key reproductive outcomes in goats. Forty crossbred multiparous goats underwent estrus synchronization and FTAI and were randomly assigned to five experimental groups (n = 8 per group): G100-7 (100 IU hCG on Day 7), G100-14 (100 IU on Day 14), G300-7 (300 IU on Day 7), G300-14 (300 IU on Day 14), and a control group (0.5 mL saline on Days 7 and 14).
Evaluated parameters included corpus luteum (CL) area, embryo implantation rate, embryonic efficiency indices (EEI-1 and EEI-2), conception rate, fertility rate, fecundity rate, fetal loss at Days 30 and 45, and total litter weight. The G300-14 group exhibited superior reproductive performance, with reduced early fetal losses, enhanced embryonic efficiency, and greater total litter weight. The authors conclude that administering 300 IU of hCG during the transitional reproductive period is a beneficial strategy for improving reproductive efficiency, particularly in low-input goat production systems.
Comment 1: Lack of Nutritional Control Data: The manuscript does not provide any information regarding dietary composition or feed intake during the period from estrus synchronization to early pregnancy. Considering that nutritional status can significantly influence luteal function and embryonic survival, it is important to clarify whether all animals were managed under uniform feeding conditions. Inclusion of such data would strengthen the validity and reproducibility of the findings.
Response to comment 1:
Text in document for comment 1: In the Comarca Lagunera, goat production is mostly managed under the sedentary extensive rangeland system, where the goats feed on the native vegetation and are allowed to forage in agricultural waste fields, mostly cantaloup, watermelon, sorghum, and oats (INIFAP, 2010).
Comment 2: Experimental Design Overlaps Significantly with Previous Literature
The experimental design is highly similar to that of a previously published study (Biology 2021, 10, 429), particularly with respect to hCG dosage, timing of administration, use of FTAI, and several key outcome measures such as CL area. Although this manuscript presents additional data—such as fetal loss rates and litter weight—the underlying mechanism investigated (i.e., enhancement of luteal function via hCG) remains essentially the same. As a result, the study lacks sufficient novelty and appears to be a confirmatory replication of existing work. The authors should clearly articulate the distinction between their findings and those of previous studies and emphasize any original contributions.
Response to comment 2: Firstly, the second study was conducted during the reproductive transition season, which in our region, Comarca Lagunera, covers the months of June and July, when sexual activity in female goats is stimulated. It is important to note that the subjects used in this study were different from those in the previous study, which was conducted during deep anestrus in March-April. The additional variables in this second study supported the higher dose of the hCG hormone, G300-14, and higher values were obtained for the remaining response variables in this experimental group. This is significant because the study was conducted during the reproductive transition season, which is a crucial period for goat farmers. The weight of the litter increases with this treatment, and we can extrapolate from this fact that in this region of Mexico, during the birthing season, more than 250,000 kids are sold at approximately 45 days of age. This study is relevant because it also reduces fetal losses at the beginning of gestation and results in a higher litter weight, as observed in this study.
Comment 3: Outdated References: Several of the references cited are relatively dated. It is recommended that the authors update their literature review with more recent studies on the application of hCG in small ruminant reproductive management to ensure the manuscript reflects current knowledge and research trends.
Response to comment 3: While it is true that some citations are outdated, these studies provide significant support for our manuscript as they are pioneering studies in the line of research addressed. It is worth noting that we will incorporate additional up-to-date citations that reflect current knowledge and research trends.
References added to the manuscript.
Rodrigues, J. N. D., Guimarães, J. D., Rangel, P. S. C., Oliveira, M. E. F., Brandão, F. Z., Bartlewski, P. M., & Fonseca, J. F. (2025). Effects of hCG administered 5 or 7 days after the onset of induced estrus on luteal morphology and function in seasonally anovular dairy goats. Animal Reproduction Science, 275, 107818.
Rodrigues, J. N. D., Guimarães, J. D., Rangel, P. S. C., Oliveira, M. E. F., Brandão, F. Z., Bartlewski, P. M., & Fonseca, J. F. (2023). Ovarian function and pregnancy rates in dairy goats that received 300 IU of human chorionic gonadotropin (hCG) intravaginally at the time of artificial insemination. Small Ruminant Research, 227, 107061.
Martins, A. L., Côrtes, L. R., Rodrigues, J. N., Rangel, P. S. C., Brandão, F. Z., Siqueira, L. G. B., ... & Fonseca, J. F. (2025). Luteal features and serum concentrations of progesterone and hCG in dairy goats submitted to estrus induction followed by intrauterine or intramuscular hCG administration. Domestic animal endocrinology, 93, 106957.
Round 2
Reviewer 3 Report
Comments and Suggestions for Authors
Regarding Comment 2: Substantial Similarity in Experimental Design to Previous Studies
Issues in the authors’ response:
Although the authors appropriately highlighted that the current study was conducted during the "reproductive transition season" — as opposed to the "deep anestrus" period examined in prior research — and included additional outcome measures such as fetal loss and litter weight, the response still falls short in articulating the fundamental scientific novelty or mechanistic insight offered by this work.
To strengthen the revised manuscript, the authors should clearly emphasize how their study provides distinct conceptual or practical advances compared to existing literature. Specifically, explicitly outlining the consistency in animal management conditions and further elaborating on the season-specific implications — particularly regarding applied outcomes such as improved litter weight and reduced fetal loss — would significantly enhance the scholarly contribution and impact of this article.
We encourage the authors to thoroughly address these aspects in their revisions.
Author Response
Comment 2: Substantial Similarity in Experimental Design to Previous Studies
Issues in the authors’ response:
Although the authors appropriately highlighted that the current study was conducted during the "reproductive transition season" — as opposed to the "deep anestrus" period examined in prior research — and included additional outcome measures such as fetal loss and litter weight, the response still falls short in articulating the fundamental scientific novelty or mechanistic insight offered by this work.
To strengthen the revised manuscript, the authors should clearly emphasize how their study provides distinct conceptual or practical advances compared to existing literature. Specifically, explicitly outlining the consistency in animal management conditions and further elaborating on the season-specific implications — particularly regarding applied outcomes such as improved litter weight and reduced fetal loss — would significantly enhance the scholarly contribution and impact of this article.
We encourage the authors to thoroughly address these aspects in their revisions.
Response 2:
In various studies cited in this article, as already reviewed, it has been noted that most studies in this region, as in other latitudes of the country and the world, only address reproductive and productive variables related to the prolificacy rate. However, as mentioned, in this study, we added the response variables of reducing fetal losses during the embryonic implantation period and total litter size. This has several significant impacts on both applied research and academic contributions. This begins with the fact that by using this reproductive control alternative, goat farmers will have higher incomes than others due to having milk and goat production during times when there is no supply and availability of these, which means that goat producers can sell their milk and goat production at a better price. Furthermore, superimposing the results obtained in this study with the total number of calves born per season in the region would yield significant results, opening areas of opportunity and addressing additional scientific questions, even with the use of lower doses of hCG.
It is worth mentioning that, because of this study, several students contributed technical support and complemented the teaching-learning process on topics related to reproductive control and its impact on the largest goat dairy basin in Mexico. This has allowed these students to complement their professional training through academic contributions.
Finally, the research project led to genetic improvement in the goat producer's herd. To date, the herd still has F1 males resulting from matings during the reproductive transition period, using reproductive biotechnology through the IATF (National Institute of Natural Resources). Therefore, this article has a valuable impact on the sedentary goat production system predominant in the Comarca Lagunera region.
This was added to the manuscript in the discussion (L418-431) as:
- It is essential to highlight that, as observed in the literature reviewed, most studies conducted both in this region [9, 10, 12, 17, 39] and globally [11, 13, 20, 29, 30, 40, 41, 46] have focused on reproductive and productive variables such as prolificacy rates. In contrast, the present study incorporates additional outcome measures, specifically the reduction of fetal losses during the embryonic implantation period and the total litter weight, thereby broadening the scope of applied and academic research impacts. The incorporation of these novel variables provides significant practical value. By employing this alternative reproductive control strategy, goat producers stand to benefit from increased incomes, particularly by ensuring milk and kid production during periods of limited supply, which in turn allows for improved market prices for both milk and kids. Moreover, when the findings of this study are considered alongside the total number of offspring born per season in the region, the results point to substantial opportunities for further inquiry—such as exploring the efficacy of lower hCG doses—and the development of targeted scientific questions.
